# Immunogenicity of Catch-Up Immunization with Conventional Inactivated Polio Vaccine among Japanese Adults

**DOI:** 10.3390/vaccines10122160

**Published:** 2022-12-15

**Authors:** Shinji Fukushima, Takashi Nakano, Hiroyuki Shimizu, Atsuo Hamada

**Affiliations:** 1Travellers’ Medical Center, Tokyo Medical University Hospital, 6-7-1 Nishishinjuku, Shinjuku-ku, Tokyo 160-0023, Japan; 2Department of Pediatrics, Kawasaki Medical School, 577 Matsushima, Kurashiki, Okayama 701-0192, Japan; 3Department of Virology II, National Institute of Infectious Diseases, 4-7-1 Gakuen, Musashimurayama-shi, Tokyo 208-0011, Japan

**Keywords:** conventional inactivated polio vaccine, trivalent oral polio vaccine, catch-up immunization, immunogenicity

## Abstract

Most Japanese adults are vaccinated twice with the Sabin trivalent oral polio vaccine. Booster vaccination is recommended for Japanese travelers to polio-endemic/high-risk countries. We assessed the catch-up immunization of healthy Japanese adults aged ≥20 years with two doses of standalone conventional inactivated polio vaccine (cIPV). Immunogenicity was evaluated by serum neutralization titers (pre-booster vaccination, 4–6 weeks after each vaccination) against type 1, 2, and 3 poliovirus strains. The participants were 61 healthy Japanese adults (26 men/35 women; mean age ± standard deviation age 35.8 ± 8.0 years). Seropositivity rates (percentage of participants with anti-poliovirus antibody titers ≥1:8) pre-vaccination were 88.5%, 95.1%, and 52.5% for Sabin strains (type 1, 2, and 3); 72.1%, 93.4%, and 31.1% for virulent poliovirus strains (type 1: Mahoney; type 2: MEF-1; and type 3: Saukett); and 93.4%, 93.4%, 93.4%, and 88.5% for type 2 vaccine-derived poliovirus strains (SV3128, SV3130, 11,196, and 11,198). After one cIPV dose, all seropositivity rates increased to 98.4–100.0%. After two cIPV doses, the seropositivity rates reached 100% for all strains. cIPV was well tolerated, with no safety concerns. Catch-up immunization with standalone cIPV induced robust immune responses in Japanese adults, indicating that one booster dose boosted serum-neutralizing antibodies to many strains.

## 1. Introduction

Since the World Health Organization (WHO) launched the global polio eradication program in 1988, the number of polio cases caused by wild polioviruses has steadily decreased worldwide from an estimate of approximately 350,000 cases in 1988 to 6 cases in 2021 [1].

However, on 5 May 2014, the WHO stated that the international spread of wild polioviruses was a Public Health Emergency of International Concern (PHEIC) [2]. As of 2022, the WHO has agreed that the risk of the international spread of poliovirus remains a PHEIC [3]. Preventing the spread of wild polioviruses and polio outbreaks caused by circulating vaccine-derived polioviruses (VDPVs) is a key challenge of polio eradication strategies [4,5,6,7].

Polio vaccines can be classified into live attenuated oral polio vaccines (OPVs) and inactivated polio vaccines (IPVs). OPVs are highly effective and safe but also offer several important advantages such as low cost, easy administration, and the induction of superior mucosal immunity compared with IPV [8,9]. 

In polio-free areas, concerns have been raised regarding two negative aspects of live attenuated OPVs. One issue is the risk of vaccine-associated paralytic poliomyelitis (VAPP) in persons receiving OPVs and those in contact with them, although this is a rare occurrence [10]. Another issue is the risk of polio outbreaks caused by the circulation of VDPVs, which are highly neurovirulent and transmissible. While OPVs are in use, VAPP and the risk of polio outbreaks caused by VDPVs are unavoidable; therefore, many polio-free countries have shifted from OPVs to IPVs [11].

In Japan, OPVs have played a leading role in the national immunization program. Routine immunization with trivalent OPV (tOPV) was initiated in 1964 [12], and standalone conventional IPV (cIPV) was approved in 2012. At present, OPV products are no longer available in Japan.

Few Japanese adults have received more than two doses of the polio vaccine because tOPV is administered twice in infancy as a routine immunization [13,14]; thus, they require catch-up immunizations by standalone cIPV when visiting polio-endemic or high-risk areas.

This study evaluated the catch-up immunization of Japanese adults with standalone cIPV.

## 2. Materials and Methods

### 2.1. Study Design and Participants

This was a descriptive study performed at Tokyo Medical University Hospital in Japan between 26 May 2011 and 31 August 2016 (UMIN000013551). The study was approved by the ethics committees of the Tokyo Medical University and the National Institute of Infectious Diseases and was conducted following the ethical principles laid out in the Declaration of Helsinki. All participants provided written informed consent before participating in the study.

Eligible participants were aged 20 years or older. Their childhood polio vaccination history was obtained from immunization records.

Exclusion criteria were pregnancy or breastfeeding, a history of poliomyelitis or polio infection, previous IPV vaccination, a history of receiving more than two doses of OPV, known or suspected congenital or acquired immunodeficiency, receipt of immunosuppressive therapy, bleeding disorders, and systemic illness.

During the study, the participants visited Tokyo Medical University Hospital three times. On their first visit, they signed an informed consent form and each participant received blood sampling and the first dose of cIPV vaccination after their eligibility was confirmed. Visit 2 was for blood sampling and a second vaccination of cIPV and Visit 3 was for blood sampling. The second and third visits occurred between 4 and 6 weeks following the previous visit.

The participants were observed for 30 min following vaccination to assess the occurrence of any immediate adverse events (AEs). They were provided with diary cards to record the solicited injection site and systemic reactions, as well as other unsolicited AEs. Solicited injection site reactions (pain, erythema, and swelling) and solicited systemic reactions (headache, malaise, myalgia, and fever) were recorded daily for 7 days post-vaccination along with any action taken to manage these AEs. Body temperature was measured daily for 7 days post-vaccination. Unsolicited AEs were recorded for 28 days post-immunization.

### 2.2. Vaccines

The study vaccine was standalone cIPV (Imovax Polio^®^, Sanofi Pasteur Inc., Swiftwater, PA, USA. Additionally, Imovax Polio^®^ subcutaneous, Sanofi K.K., Tokyo, Japan). In 2011–2012, we imported Imovax Polio^®^ for this study. In 2012, Imovax Polio^®^ for subcutaneous use was approved in Japan, so this vaccine was used in this study. Each participant received ‘Imovax Polio^®^’ intramuscularly in the deltoid region or ‘Imovax Polio^®^ for subcutaneous use’ subcutaneously in the upper arm region.

Each vaccine was a trivalent IPV provided in a pre-filled syringe with a needle representing a single dose (0.5 mL). The vaccine contained three types of inactivated poliovirus D-antigens: 40 DU type 1 (Mahoney strain), 8 DU type 2 (MEF-l strain), and 32 DU type 3 (Saukett strain). In addition, 2-phenoxyethanol and formaldehyde were included as preservatives. The vaccines were stored at 2–8 °C.

### 2.3. Immunogenicity Evaluation and Serological Analysis

Blood samples (6 mL) were collected in dry, sterile, capped plastic tubes for the assessment of neutralizing antibody titers against polioviruses prior to and 4–6 weeks post-vaccination. The blood was allowed to clot, and serum was separated by centrifuging at 1500× *g* for 10 min. The serum samples were stored at ≤−20 °C. 

Immunogenicity was assessed by measuring serum neutralization titers against type 1, 2, and 3 polioviruses in HEp-2 cells using a microneutralization assay. Neutralizing antibody titers were measured at the National Institute of Infectious Diseases (Tokyo, Japan). Viruses included Sabin strains (type 1, 2, and 3), virulent poliovirus strains (type 1: Mahoney strain; type 2: MEF-1 strain; and type 3: Saukett strain), and type 2 VDPVs derived from sporadic cases of acute flaccid paralysis in Vietnam in 2012 (SV3128 and SV3130) and cases from an outbreak in Nigeria in 2005 (11,196 and 11,198).

VDPVs are OPV-derived variants with more than 1% nucleotide divergence in the VP1 protein from Sabin 1 and Sabin 3 strains, and more than 0.6% nucleotide divergence from the Sabin 2 strain [15]. VDPVs are further classified into circulating VDPVs (cVDPV) identified in community transmission [16,17], immunodeficient VDPVs (iVDPV) from cases with primary immunodeficiency [18], and ambiguous VDPVs (aVDPVs) that have no evidence of cVDPV or iVDPV cases.

The titer of neutralizing antibody required for protection against poliovirus was assumed to be 1:8 (1/dil) or higher [19]. Neutralizing antibody titers below the threshold of 1:4 were assigned a value of 1:4, and those above a threshold of 1:1024 were assigned a value of 1:1024. 

### 2.4. Safety

Safety endpoints included occurrence, nature, duration, intensity, and the relationship to vaccination of unsolicited systemic AEs reported within 30 min; occurrence, time to onset, duration, and intensity of pre-defined (solicited) injection site reactions and systemic reactions occurring from day 0 to day 7; occurrence, nature, time to onset, duration, maximum intensity (for non-serious AEs only), and causal relationship to vaccination (for systemic AEs only) of unsolicited AEs within 28 days; and occurrence, nature, time to onset, duration, relationship to vaccination, and outcome of serious AEs (SAEs) occurring for the entire study duration. 

Pain at the injection site was graded as follows: grade 1, no interference with activity; grade 2, some interference with activity; and grade 3, significant interference and prevention of daily activity. Erythema and swelling at the injection site were graded as follows: grade 1, ≥25 to ≤50 mm; grade 2, ≥51 to ≤100 mm; and grade 3, >100 mm. In all participants, solicited systemic reactions were graded as follows: Fever—grade 1, ≥37.5 °C to ≤38.4 °C; grade 2, ≥38.5 °C to ≤38.9 °C; and grade 3, ≥39.0 °C. Headache, malaise, and myalgia were graded as follows: grade 1, no interference with activity; grade 2, some interference with activity; and grade 3, significant interference and prevention of activity. 

Unsolicited adverse events were AEs that did not fulfill the conditions prelisted in the report form either in terms of symptoms or time of onset post-vaccination. Adverse reactions (ARs) were AEs in which a causal relationship between the vaccine and the AE was at least a reasonable possibility. For all safety endpoints, the proportion of participants experiencing at least one event were determined.

### 2.5. Statistical Analysis

To assess the immunogenicity of polio vaccines, the seropositivity rate, geometric mean titers (GMT), and geometric mean fold rise (GMFR) were calculated. The seropositivity rates, GMTs, and 95% confidence intervals (CIs) were calculated at Visit 1 (pre-vaccination) and at Visits 2 (post-vaccination with the first dose of cIPV) and 3 (post-vaccination with the second dose of cIPV). Seropositivity was defined as the percentage of participants with neutralizing antibody titers of ≥1:8. The exact binomial distribution (Clopper–Pearson method) was used to calculate the CI for proportions. The 95% CIs of the GMT point estimates were calculated using normal approximation, assuming they were log-normally distributed. The significance of GMT within each strain group was assessed by the Wilcoxon signed-rank test. GMFR is the geometric mean of the ratios of post-dose antibody to pre-dose antibody. For safety evaluation, the number and percentage of participants who experienced at least one AE were calculated. All statistical analyses were performed with EZR [20], which is a modified version of R commander designed to add the statistical functions frequently used in biostatistics.

## 3. Results

### 3.1. Baseline Characteristics of Participants

Sixty-two participants were enrolled in this study, but one participant dropped out. In total, 61 adults participated, and their characteristics are shown in Table 1. The 61 participants comprised 26 men and 35 women with a mean ± standard deviation age of 35.8 ± 8.0 years (range: 20–57).

Thirty-seven participants had received two doses of primary tOPV in childhood, and two participants had received one dose of tOPV. One participant was unvaccinated with tOPV.

The number of tOPV doses for twenty-one participants was unknown because they had not kept their immunization records.

### 3.2. Immunogenicity

Forty-nine participants received ‘Imovax Polio^®^’ intramuscularly and twelve participants received ‘Imovax Polio^®^ for subcutaneous use’ subcutaneously.

Before vaccination, the seropositivity rates (defined as neutralization titers ≥1:8) were as follows: Sabin 1 (88.5%), Sabin 2 (95.1%), Sabin 3 (52.5%), Mahoney (72.1%), MEF-1 (93.4%), Saukett (31.1%), SV3128 (93.4%), SV3130 (93.4%), 11196 (93.4%), and 11198 (88.5%). The seropositivity rates after the first dose were as follows: Sabin 1 (98.4%), Sabin 2 (100%), Sabin 3 (98.4%), Mahoney (98.4%), MEF-1 (100%), Saukett (98.4%), SV3128 (100%), SV3130 (100%), 11196 (100%), and 11198 (100%). Only one participant did not achieve a protective level of neutralizing antibodies against poliovirus strains (Sabin 1, Sabin 3, Mahoney, and Saukett) after the first dose, but this individual had not received tOPV in childhood. The second dose resulted in 100% seropositivity rates against all poliovirus strains tested (Table 2). 

Table 3 shows the GMTs of anti-poliovirus antibody. For the Sabin strains (types 1, 2, and 3), the GMTs increased from 42.0, 44.5, and 10.4 pre-vaccination to 745, 914, and 561 after the first dose, and to 737, 883, and 536 after the second dose, respectively. For virulent poliovirus strains (type 1: Mahoney strain; type 2: MEF-1 strain; and type 3: Saukett strain), the GMTs increased from 14.9, 40.2, and 6.9 pre-vaccination to 643, 924, and 495 after the first dose, and to 621, 914, and 478 after the second dose, respectively. For type 2 VDPVs (SV3128, SV3130, 11196, and 11198), the GMTs increased from 55.2, 37.5, 42.0, and 24.9 pre-vaccination to 957, 935, 904, and 924 after the first dose, and to 957, 946, 957, and 864 after the second dose, respectively.

There was no significant difference between the GMTs after the first or second dose.

Even one booster dose with cIPV was effective at enhancing antibody titers in two participants who had received one dose of tOPV, but one participant who was unvaccinated with tOPV needed two doses of cIPV for seropositive titer (Table 4).

### 3.3. Safety

We received the 116 diary cards from the participants. A summary of the safety profile of cIPV is shown in Table 5. No immediate unsolicited systemic AEs, SAEs, or deaths were reported. Solicited injection site reactions occurred in 42 participants (36.2%), and solicited systemic reactions occurred in 13 participants (11.2%).

The most common solicited injection site reactions were pain (*n* = 40), swelling (*n* = 15), and erythema (*n* = 8). All injection site reactions were either grade 1 or grade 2. The most common solicited systemic reactions were malaise (*n* = 9), headache (*n* = 5), and myalgia (*n* = 2). Fever was not reported. All systemic reactions were either grade 1 or grade 2.

Unsolicited AEs were reported in six participants and included upper respiratory tract infection (*n* = 5), diarrhea (*n* = 1), and influenza type A infection (*n* = 1). These AEs were unrelated to vaccination with cIPV.

## 4. Discussion

Worldwide, the common knowledge is that most adults do not need polio booster vaccination if they were previously vaccinated as a part of routine immunization schedules [21,22]. However, adults at higher risk should consider polio vaccination if they fall into the following groups: (i) travelers to polio-endemic or high-risk areas, (ii) workers in facilities that handle poliovirus infectious materials, and (iii) healthcare workers treating patients who might have polio or be in close contact with poliovirus-infected individuals. Adults with these criteria who have previously received one or two doses of poliovirus vaccine should receive an additional one or two doses. The time elapsed since the earlier dose(s) is immaterial.

As of September 2012 in Japan, all infants began to be immunized with IPV (either stand-alone or in combination with diphtheria, tetanus, and acellular pertussis vaccines), rather than tOPV from the Sabin strain. In the current schedule, four doses of IPV-containing vaccines are administered.

Globally, routine polio immunization schedules involve three or more doses, whereas most Japanese adults received only two doses of tOPV in childhood according to the national immunization program in Japan. As such, Japanese adults can be considered incompletely immunized (previously received only one or two doses of tOPV) according to the definition of the global OPV standards [23].

In Japan, polio booster vaccination is not recommended for adults who have received two doses of tOPV. However, Japanese adult travelers entering polio-endemic or high-risk areas should receive one or two doses with cIPV.

In our study, we found that polio neutralization titers before cIPV vaccination in most Japanese adults were maintained at protective levels, except for type 3 poliovirus. This result shows that tOPV is effective at conferring durable protection against paralytic poliomyelitis. As passively acquired booster effects resulting from natural contact with circulating polioviruses are limited in Japan, this suggests that most Japanese adults vaccinated against polio during infancy will maintain life-long immunity against the virus. However, neutralizing antibody titers against the Sabin 3 and Saukett strains were relatively low in this study. This trend is consistent with previous seroepidemiological data reported in Japan [24,25], presumably due to interference of the intestinal viral replication of Sabin 3 by Sabin 2 during the initial tOPV doses [26]. Although the response to type 3 poliovirus was generally lower than those to the other serotypes after tOPV immunization in OPV-using developing countries, the increasing number of tOPV doses was associated with a higher seroprevalence to type 3 poliovirus [27,28].

Following catch-up immunization, the antibody titers significantly increased and reached a protective level against all strains. Only one participant did not achieve a protective level of neutralizing antibodies against poliovirus strains (Sabin 1, Sabin 3, Mahoney, and Saukett) after the first vaccination, but this individual had not received tOPV in childhood. Even one booster dose with cIPV was effective at enhancing antibody titers in two adults who had received one dose of tOPV. Other studies have suggested that individual-level immunity may be better maintained when a primary OPV immunization is boosted by IPV [29,30]. Therefore, unvaccinated adults who are at increased risk of exposure to poliovirus should be given a total of three doses of cIPV at the recommended intervals (0, 1–2, 6–12 months). Adults who are incompletely vaccinated should receive the remaining doses of IPV to complete the three-dose series [21].

In our study, neutralizing antibodies were efficiently induced regardless of the route of injection, and seroconversion did not depend on age group or sex. Our results support the idea that one booster dose with cIPV is sufficient to boost serum-neutralizing antibodies to a wide range of strains.

During the study, no immediate systemic AEs were reported. Overall, 36.2% of participants experienced at least one solicited injection site AE (e.g., erythema, swelling, or pain). Moreover, 11.2% of participants experienced at least one solicited systemic AE (e.g., malaise, fever, headache, or myalgia). Unsolicited AEs were reported in six participants and included upper respiratory tract infection (five participants), diarrhea (one participant), and influenza A infection (one participant). These AEs were unrelated to vaccination with cIPV.

Several limitations of our study need to be considered before making any generalizations. First, this was a single-site study with a small sample size, and the number of males and females was not balanced. Nevertheless, poliovirus antibodies were significantly increased in all participants following cIPV booster vaccination. Second, the sample size was too small to address the safety concerns.

In summary, this study found that catch-up immunization with cIPV is safe and immunogenic in previously tOPV-immunized adults. The results support the idea that one booster dose is sufficient to boost circulating neutralizing antibodies to a wide range of strains among adults incompletely vaccinated against polio. The results of this study can be applied to people in developing countries who receive less than the prescribed number of oral polio vaccinations.

## Figures and Tables

**Table 1 vaccines-10-02160-t001:** Baseline characteristics of study participants.

Characteristics		N (%)
Sex		
	Female	35 (57)
	Male	26 (43)
Age (years)		
	20–29	13 (21)
	30–39	33 (54)
	40–49	11 (18)
	50–59	4 (7)
Doses of primary trivalent oral polio vaccine (tOPV) in childhood		
	0	1 (2)
	1	2 (3)
	2	37 (61)
	unknown	21 (34)

N = 61, number of participants completed study protocol.

**Table 2 vaccines-10-02160-t002:** Seropositivity rate against poliovirus strains.

Poliovirus Strain		Visit 1			Visit 2			Visit 3		
		n/N	%	(95%CI)	n/N	%	(95%CI)	n/N	%	(95%CI)
Sabin	Sabin 1	54/61	88.5	(77.8–95.3)	60/61	98.4	(91.2–100)	61/61	100	(94.1–100)
	Sabin 2	58/61	95.1	(86.3–99.0)	61/61	100	(94.1–100)	61/61	100	(94.1–100)
	Sabin 3	32/61	52.5	(39.3–65.4)	60/61	98.4	(91.2–100)	61/61	100	(94.1–100)
Virulent	Mahoney (type 1)	44/61	72.1	(59.2–82.9)	60/61	98.4	(91.2–100)	61/61	100	(94.1–100)
	MEF-1 (type 2)	57/61	93.4	(84.1–98.2)	61/61	100	(94.1–100)	61/61	100	(94.1–100)
	Saukett (type 3)	19/61	31.1	(19.9–44.3)	60/61	98.4	(91.2–100)	61/61	100	(94.1–100)
Type 2 vaccine-derived polioviruses (VDPV)	SV3128	57/61	93.4	(84.1–98.2)	61/61	100	(94.1–100)	61/61	100	(94.1–100)
	SV3130	57/61	93.4	(84.1–98.3)	61/61	100	(94.1–100)	61/61	100	(94.1–100)
	11196	57/61	93.4	(84.1–98.4)	61/61	100	(94.1–100)	61/61	100	(94.1–100)
	11198	54/61	88.5	(77.8–95.3)	61/61	100	(94.1–100)	61/61	100	(94.1–100)

n, number of participants with seropositive titer; N, number of participants with valid serology data; CI, confidence interval.

**Table 3 vaccines-10-02160-t003:** Geometric mean titers (GMTs) against poliovirus strains.

Poliovirus Strain		GMT (95%CI)						GMFR		GMFR	
		Visit 1		Visit 2		Visit 3		Visit 2/Visit 1		Visit 3/Visit 2	
Sabin	Sabin 1	42.0	(28.6–60.8)	745	(582–944)	737	(603–912)	17.7	(*p* < 0.01)	1.0	(*p* = 0.83)
	Sabin 2	44.5	(32.7–61.1)	914	(805–1033)	883	(778–1021)	20.5	(*p* < 0.01)	1.0	(*p* = 0.233)
	Sabin 3	10.4	(7.57–14.5)	561	(411–769)	536	(409–705)	53.9	(*p* < 0.01)	1.0	(*p* = 0.148)
Virulent	Mahoney (type 1)	14.9	(10.9–20.0)	643	(486–818)	621	(495–767)	43.1	(*p* < 0.01)	1.0	(*p* = 0.193)
	MEF-1 (type 2)	40.2	(30.2–52.5)	924	(834–1045)	914	(824–1009)	23.0	(*p* < 0.01)	1.0	(*p* = 0.85)
	Saukett (type 3)	6.90	(5.40–8.87)	495	(352–681)	478	(356–643)	71.7	(*p* < 0.01)	1.0	(*p* = 0.606)
Type 2 VDPV	SV3128	55.2	(40.9–73.8)	957	(843–1081)	957	(869–1050)	17.3	(*p* < 0.01)	1.0	(*p* = 1.000)
	SV3130	37.5	(27.7–49.9)	935	(824–1057)	946	(849–1074)	24.9	(*p* < 0.01)	1.0	(*p* = 0.773)
	11196	42.0	(31.0–56.0)	904	(802–1038)	957	(851–1072)	21.5	(*p* < 0.01)	1.1	(*p* = 0.037)
	11198	24.9	(18.5–34.1)	924	(830–1050)	864	(760–998)	37.1	(*p* < 0.01)	0.9	(*p* = 0.047)

GMT, geometric mean titer; CI, confidence interval; GMFR, geometric mean fold rise, the geometric mean of the ratios of post-dose antibody to pre-dose antibody; *p*-values were calculated using the Wilcoxon signed-rank test.

**Table 4 vaccines-10-02160-t004:** Antibody titers of three participants who had received only one or no dose of trivalent OPV.

Poliovirus Strain		Participant 1 (One Dose with tOPV)			Participant 2 (One Dose with tOPV)			Participant 3 (Unvaccinated with tOPV)		
		Visit 1	Visit 2	Visit 3	Visit 1	Visit 2	Visit 3	Visit 1	Visit 2	Visit 3
	Sabin 1	128	>1024	>1024	8	>1024	>1024	<4	<4	16
Sabin	Sabin 2	128	>1024	>1024	8	512	512	<4	32	32
	Sabin 3	<4	64	128	8	>1024	>1024	<4	<4	128
	Mahoney (type 1)	8	>1024	>1024	4	1024	1024	<4	4	128
Virulent	MEF-1 (type 2)	64	>1024	>1024	4	>1024	>1024	<4	64	128
	Saukett (type 3)	<4	64	256	4	>1024	>1024	<4	<4	64
Type 2 VDPV	SV3128	64	>1024	>1024	4	>1024	1024	<4	32	64
	SV3130	64	>1024	>1024	4	>1024	>1024	<4	32	32
	11196	64	>1024	>1024	4	>1024	1024	<4	32	32
	11198	64	>1024	>1024	4	512	512	<4	32	32

Participants 1 and 2 had received one dose of tOPV, but participant 3 was unvaccinated with tOPV in childhood. Even one booster dose with cIPV was effective at enhancing antibody titers in participants 1 and 2, but participant 3 needed two doses of cIPV for a seropositive titer against all strains.

**Table 5 vaccines-10-02160-t005:** Summary of safety data.

Symptoms	Grade of Seriousness	n/N (%)
Immediate AEs		0/122 (0.0)
Solicited Injection Site Reaction		42/116 (36.2)
Pain		40/116 (34.5)
	Grade 1	39/116 (33.6)
	Grade 2	1/116 (0.9)
	Grade 3	0/116 (0.0)
Swelling		15/116 (12.9)
	Grade 1	14/116 (12.1)
	Grade 2	1/116 (0.9)
	Grade 3	0/116 (0.0)
Erythema		8/116 (6.9)
	Grade 1	8/116 (6.9)
	Grade 2	0/116 (0.0)
	Grade 3	0/116 (0.0)
Solicited Systemic Reaction		13/116 (11.2)
Fever		0/116 (0.0)
	Grade 1	0/116 (0.0)
	Grade 2	0/116 (0.0)
	Grade 3	0/116 (0.0)
Malaise		9/116 (7.8)
	Grade 1	8/116 (6.9)
	Grade 2	1/116 (0.9)
	Grade 3	0/116 (0.0)
Headache		5/116 (4.3)
	Grade 1	5/116 (4.3)
	Grade 2	0/116 (0.0)
	Grade 3	0/116 (0.0)
Myalgia		2/116 (1.7)
	Grade 1	2/116 (1.7)
	Grade 2	0/116 (0.0)
	Grade 3	0/116 (0.0)
Serious AEs		0/116 (0.0)
Death		0/116 (0.0)

n, number of participants reporting at least one event or reaction; N, number of participants with available data; AEs, adverse events.

## Data Availability

Not applicable.

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
