# Peer review of "Immunogenicity of Catch-Up Immunization with Conventional Inactivated Polio Vaccine among Japanese Adults"

_vaccines, 2022, doi:10.3390/vaccines10122160_

Round 1

Reviewer 1 Report

The article by Fukushima et al. reports data regarding the seroconversion of individuals vaccinated with the inactivated polio vaccine Japan. Although the number of people is quite low, this paper provides valuable data that are of interest for experts in the field. Please find my comments and suggestions below.

1-    Table 2 shows a lower seropositivity rate against PV3 compared to the two other serotypes. In the Discussion section, the authors indicate that similar observation were already reported Japan (lines 237-238). Is this specific to Japan or was it already observed in other contexts? Do the authors have hypothesis about the causes of this observation?

2-    Table 4 shows that Participant 3 had no antibody at the 1st visit time and that two rounds of vaccination induced PV antibody titers substantially lower than those observed for Participants 1 & 2. What can explain this relatively limited seroconversion? Does this participant have peculiar treats that could explain this (age? Immune disease?)?

3-    In my opinion, Table 1 could be replaced by histograms that would be easier to read.

Author Response

Thank you for reviewing our manuscript.

We have revised the manuscript according to your commnts.

Reviewer 2 Report

    In this manuscript, Fukushima et al. studied the catch-up immunization of Japanese adults with two doses of the conventional inactivated polio vaccine (cIPV). This immunogenicity study was performed at Tokyo Medical University Hospital between May 2011 and August 2016. The participants were recruited from 61 adults aged ³20 years, including 35 women and 26 men. Based on their polio vaccination records, most of them were administrated with two doses of the trivalent oral polio vaccine (tOPV) in infancy. In the study, seropositivity rates and geometric mean titers against various of poliovirus strains were assessed and measured. The viruses included Sabin strains (types 1, 2, and 3), virulent poliovirus strains (type 1: Mahoney strain, type 2: MEF-1 strain, and type 3: Saukett strain), and type-2 VDPVs from sporadic cases in Vietnam in 2012 (SV3128 and SV3130), and outbreak cases in Nigeria in 2005 (11196 and 11198). It was found that all seropositivity rates of participants increased to >98.4% after they received one cIPV dose and reached 100% after two cIPV doses. Moreover, cIPV was found to be well tolerated to the participants. Thus booster vaccination is recommended for Japanese adults when they plan to travel to the polio-endemic or high-risk areas.

      Overall, the manuscript provides valuable information to the global polio eradication program. Quality and importance of this manuscript meet the standard of Vaccines for its publication. However, some major points particularly related to the tables have to be improved and clarified before it is accepted.

1.  The presentation of Table 1 is unclear and suggested to contain three or more columns. For example, the first column therein is mixed with the main characteristics (Sex, Age group, Doses of primary oral polio vaccine in childhood, and Route of vaccination) and their sub characteristics, such as Female and Male for Sex (or Gender) and others. It is suggested to divide them into two columns. Moreover, Female and Male at the second column are redundant and should be deleted. In addition, the “Mean age ± Standard deviation (years)” and “Minimum:Maximum” do not have the %. These two rows are suggested to list at the footnote. Moreover, what is “Minimum:Maximum”? Is “Route of vaccination” about the primary tOPV or cIPV? If it is the latter, the “Route of vaccination” does not belong to the baseline characteristics of study participants.

 2.  For Tables 2−4, all of the top row tabular headers regarding to the viruses have to be provided.

 3.  In Table 3, 42 should be 42.x and 6.9 is 6.9x as there are three significant figures for others. All of the numbers for (95%CI) need to be corrected; the “;” should be changed to “−“. How to calculate the “Visit 2/1” and “Visit 3/2” should be indicated at the footnote. Moreover, “Visit 2/1” is suggested to change to “Visit 2/Visit 1” so as to “Visit 3/2”. The interpretation of “Rise fold” shown in the Table 3 should be described in the main text. “Rise fold*” needs to be centered between Visit 2/1 and Visit 3/2.

4.  More description in the main text should be added for the data shown in Table 4 .

5. It is inappropriate for the Table 5 which includes only two columns. Thus it should be revised as the suggestion provided for Table 1. Furthermore, the presentation about n/N (%) is unclear and has to be revised. Why 122, 116, or 30 are associated with N and some rows do not have N? The definition of N has to be provided. Additionally, the footnote is suggested to add on the data for various of grade 1 and grade 2 as they represent different evaluation of symptoms.

6. The keyword “safety” is suggested to be replaced by other keyword(s) that is related to analysis or evaluation methods in this manuscript.

Author Response

Thank you for reviewing our manuscript.

We have revised the manuscript according to your comments.

Round 2

Reviewer 2 Report

In this revised manuscript, the authors have answered all the questions raised by the reviewer point-by-point. However, there are still some minor points regarding to Tables, which need to be improved to allow the readers to follow easily.

1.  Add the top row tabular headers, such as characteristics or factors, in Table 1. In addition, break the “Doses of primary …. in childhood” into two lines and add the definition of N at the footnote.

2.   Delete “Confidence Interval” from “95% Confidence Interval (CI)” as it has been defined at the footnote. Give spaces between “n/N%” for Visit 2 and Visit 3; align them with their belonged columns.

3.  Add a space between Visit2 (the last word at the row 2) and revise “GMTR is the geometric ….” to “GMTR, geometric ….”

4.  Add the top row tabular headers, such as symtoms and level (or grade) of seriousness.

After fulfilling these editing works by authors, the revised version will be recommended for publication in Vaccines.

Author Response

Thank you very much for reviewing our manuscript.

We have revised the Tables according to your comments.

Please see the attachment. We highlighted the revised parts in yellow
